# Characterization of the Germplasm Bank for the Spanish Autochthonous Bull Breed “Asturiana de la Montaña”

**DOI:** 10.3390/ani13081402

**Published:** 2023-04-19

**Authors:** Carolina Tamargo, Amer Salman, Jose Néstor Caamaño, Felipe Martínez-Pastor, Ángel Fernández, Rodrigo Muiño, María Teresa Carbajo, Carlos Olegario Hidalgo

**Affiliations:** 1Selección y Reproducción Animal—SERIDA, Principado de Asturias, 33394 Gijón, Spain; 2INDEGSAL and IMAPOR, Universidad de León, 24071 León, Spain; 3Department of Molecular Biology (Cell Biology), Universidad de León, 24071 León, Spain; 4Department of Animal Patology, Universidad de Santiago de Compostela, 15705 Santiago de Compostela, Spain; 5Department of Medicine, Surgery and Veterinary Anatomy, Universidad de León, 24071 León, Spain

**Keywords:** cattle, Asturiana de la Montaña, semen bank, semen cryopreservation, fertility

## Abstract

**Simple Summary:**

Germplasm banks provide an essential service by safeguarding the genetics of endangered or valuable species or individuals. The SERIDA (Asturias, Spain) keeps germplasm from several valuable local breeds. This study aimed to characterize the viability of the cryopreserved semen samples from the “Asturiana de la Montaña,” a valuable autochthonous breed adapted to the mountainous Atlantic environment. We assessed the sperm quality of the semen doses from 40 bulls after thawing. We combined this data with fresh semen quality and fertility data from females inseminated with semen doses from these bulls. Results showed that “Asturiana de la Montaña” bulls produce semen of comparable quality to commercial breeds, fresh and frozen-thawed, confirming the viability and usefulness of the SERIDA germplasm for preserving and spreading this breed’s genetics. The fertility of the semen doses was also acceptable. However, we did not find a relationship between the semen quality results and the field fertility, except for the linearity of the spermatozoa (proportion swimming in more linear paths). Whereas the sperm parameters of this endangered breed are acceptable, future research should focus on optimizing the existing protocols for further improving post-thawing semen quality.

**Abstract:**

Semen cryobanks are critical for preserving autochthonous and rare breeds. Since sperm cryopreservation has been optimized for commercial breeds, non-commercial ones (often endangered) must be characterized to ensure the germplasm’s viability. This study reports an investigation of the “Asturiana de la Montaña” breed (AM), a valuable Spanish autochthonous cattle breed adapted to the mountainous Atlantic environment. The survey included cryopreserved semen doses from 40 bulls stored at the Principado de Asturias Germplasm Bank. Data were obtained from the routine fresh semen analysis, CASA (motility), and flow cytometry analyses of fresh and post-thawing semen, and the 56-day non-return-rate (NRR) in heifers and cows (all results as 1st and 3rd quartiles). Fresh samples (artificial vagina) were within the normal range for cattle (4–6 mL, 5–10 × 10^9^/mL; mass motility 5). Post-thawing results showed motility below typical for commercial breeds (total motility 26–43%, progressive 14–28%), with higher values for viability (47–62%). Insemination results showed a good performance for this breed (NRR: 47–56%; higher for heifers). Sperm volume increased with age, with little or no effects on sperm quality. Few associations were found between post-thawing quality or freezability and NRR, LIN being the variable more strongly associated (positively). The AM semen bank shows a good prospect for preserving and disseminating the genetics of this breed. This survey indicates that dedicated research is needed to adapt freezing protocols to this breed, optimizing post-thawing results.

## 1. Introduction

Artificial insemination with cryopreserved semen (AI) is the practice with the highest impact on improving animal production [1,2]. AI is still the most widespread and a cornerstone in dairy and beef cattle, a routine method in the former and a tool for increasing genetics in the latter [3]. The impact of AI would not have been possible without the parallel development of semen cryopreservation and the successful distribution of frozen doses. Indeed, 95% of inseminations in dairy cattle come from AI with deep-frozen semen [4], and the importance in beef cattle in increasing [5]. Current methods have allowed good post-thawing fertility of bull spermatozoa in different scenarios, but between-male variability and declining cow fertility due to the selection for milk productivity are still important challenges [6].

These technologies have been highly successful in commercial breeds due to intense research and efforts in their application and coordination with genetics programs. In contrast, the situation has been different for autochthonous or rare breeds, which commercial ones have gradually displaced. Fortunately, there has been a renewed interest in these breeds, not only due to their cultural and social importance but also because of specific characteristics (e.g., rusticity, adaptation to particular environments) of great interest in a world facing environmental challenges [7,8,9].

Germplasm banks are one of the strategies to preserve endangered domestic breeds [10,11]. However, these breeds have a very different context and history than commercial ones regarding genetics selection, inbreeding, farm modernization, and rearing methods. Considering semen cryopreservation and AI, the protocols adapted for commercial breeds might not be optimal, and therefore they must be tested and improved if necessary. Doing otherwise could jeopardize the future use of germplasm for these valuable breeds.

In this study, we carried out a retrospective study on the Principado de Asturias semen bank (Centro de Biotecnología Animal CBA, Deva- SERIDA) for the “Asturiana de la Montaña” (AM) cattle breed. This germplasm bank was established in 2004 and currently stores semen for many breeder associations of autochthonous breeds [12]. AM is considered an endangered breed and is officially protected (National Official Breed Catalog). It belongs to the “Cantabric trunk” of Iberian bovine breeds, and the local population has used it from time immemorial. The main uses are forest and pasture-keeping and meat production. This breed is especially valued for maintaining the local culture and economic activity.

Previously, we explored the freezing extender’s effects on this breed’s post-thawing quality [13]. In this report, we focused on characterizing the sperm quality before and after cryopreservation, and the AI results in the field obtained by the breeders from the stored doses. We also considered the post-thaw cryotolerance of spermatozoa among individual males. AI practitioners using cryopreserved semen in many species have identified individual males whose spermatozoa either consistently fail to survive the freeze-thaw process or survive particularly well, raising the terminology for “good” and “bad freezers” [14,15]. The differences between the individuals within the same animal species in the cryosurvival of spermatozoa and the susceptibility of spermatozoa to cryoinjury could be genetic. Since autochthonous breeds have not been submitted to the strict selection of commercial breeds, this variability may be higher. It is necessary to characterize the post-thawing quality of these breeds to aim future studies at improving freezing protocols for specific males according to the freezability of their semen.

Therefore, we set two objectives for this study. The first one was carrying out a survey on the post-thawing quality of the “Asturiana de la Montaña” semen bank, considering the post-thawing quality data together with the recorded pre-freezing one, also comparing these results with those from commercial breeds to assess the viability of the bank. Our hypothesis here was that our results for AM would be acceptable if even slightly inferior to those reported for those commercial breeds, since protocols have not been specifically adapted yet. In addition, as a second objective, we aimed at enhancing our knowledge on the spermatology and reproductive performance of this breed, by including information not only on the sperm quality before and after freezing/thawing, but also considering fertility data from the genetics dissemination program of the breeders’ association (ASEAMO). Whereas this second objective was predefined as limited (since only non-return rates, NRR, are currently available), we considered that the information could be relevant both for the conservation of the breed and for defining future, more specific, studies.

## 2. Material and Methods

### 2.1. Animals, Semen Collection, Evaluation, and Freezing

All the procedures involving the animals (bulls for sperm collection and cows for AI) were carried out by trained staff and veterinarians in the SERIDA and the breeders’ associations (ASEAMO —Spanish Association for Select Asturiana de la Montaña Cattle— and ASCOL—Asturian Holstein Breeders’ Association), following Spanish Regulations for farm animals’ management (Law 6/2013). The present study was carried out with semen doses provided by the breeding center for research, and therefore the authors did not manage animals in this study. The experimental plan was reviewed by the Animal Care Committee of the University of León, which cleared it as having no concern regarding animal welfare.

The germplasm bank is located at the CBA-SERIDA (Deva, Asturias, Spain). The survey included semen doses from 40 AM bulls (1 to 4 ejaculates each), collected from 1997 to 2015. The median age was 28.4 months (interquartile: 25.8, 32.2). The doses stored in the bank were prepared as follows. Semen was collected by an artificial vagina (45 °C) and kept in a water bath at 30 °C. Volume was estimated from the graduated tube. An aliquot was extended in a citrate buffer, and the concentration was estimated by absorbance with an adapted photometer calibrated for bull semen (Accucell, IMV Technologies, L’Aigle, France). Mass motility (0–5 scale) was subjectively estimated in a semen drop at low magnification (×4 objective) with a phase contrast microscope and heated stage (37 °C). Only samples with mass motility ≥4 were processed for cryopreservation. Semen was extended with Biociphos (15 bulls) or BIOXcell (25 bulls) to a final sperm concentration of 92 × 10^6^ mL^−1^. Both extenders were acquired from IMV (L’Aigle, France), and BIOXcell replaced Biociphos when the manufacturer discontinued the latter by 2006. After the extension, semen doses were cooled to 4 °C for 90 min, equilibrated for 3 h, and packaged in 0.25 mL plastic French straws. Straws were frozen in a programmable freezer (Digit-cool; IMV Technologies) with the curve [16]: −5 °C/min from +4 to −10 °C; −40 °C/min from −10 to −100 °C and −20 °C/min from −100 to −140 °C. Then, samples were stored in liquid nitrogen in the germplasm bank.

Thawing was performed in a water bath at 37 °C for 30 s before analyses. Three straws per ejaculate were simultaneously thawed and pooled in a pre-warmed 1.5 mL tube and analyzed after 10 min at 25 °C in the dark.

### 2.2. CASA Analysis

An aliquot of each sample was diluted in PBS and observed in an ISAS chamber (Proiser, Paterna, Spain) with a negative phase contrast microscope with a warming stage at 37 °C. At least 5 fields and 200 cells were recorded at ×200, at 25 frames/s for 1 s, with a Basler A302fs digital camera (Basler Vision Technologies, Ahrensburg, Germany). The images were processed with the ISAS software (Proiser). In this study, we show parameters: Total motility (MOT), progressive motility (PROG, VAP > 25 µm/s and STR > 80%), VCL (curvilinear path velocity), VSL (straight path velocity), VAP (average path velocity according to the average—smoothed—path; µm/s), LIN (linearity), STR (straightness), WOB (wobble), ALH (amplitude of the lateral displacement of the sperm head), and BCF (frequency of the flagellar beat).

### 2.3. Abnormal Forms and Cytoplasmic Droplets Assessment

We determined the percentage of abnormal spermatozoa and cytoplasmic droplets by fixing 5 µL of sample in 500 µL of buffered 2% glutaraldehyde. At least 200 spermatozoa were assessed at ×400 phase contrast.

### 2.4. Hypoosmotic Swelling Test (HOS Test)

The functional integrity of the sperm plasma membrane was assessed by the HOS test. Five microliters of sample were incubated in 500 µL of sodium citrate solution at 100 mOsm/kg at room temperature, then adding 5 µL of commercial glutaraldehyde for fixing. At least 200 spermatozoa were examined at ×400 phase contrast, considering those with a swollen tail as positive.

### 2.5. Flow Cytometry Analyses

We used flow cytometry to assess sperm viability and acrosomal status. Fluorescent probes and cytometry tubes were purchased from Thermo Fisher Scientific (Waltham, MA, USA), and cytometry consumables from Beckman Coulter (Brea, CA, USA). The staining solutions were prepared the same day in PBS with PBS 0.5% BSA (bovine serum albumin) with propidium iodide (PI, 3 µM) for evaluating membrane integrity and PNA-FITC (peanut agglutinin, 1 µg/mL) for acrosomal status. The semen samples were added at 1 × 10^6^ mL^−1^ and incubated for 15 min at 37 °C. Analyses were performed with a CyAn ADP cytometer (Beckman Coulter, Brea, CA, USA). Excitation was provided with a 488 nm laser, and the fluorescence was collected by photodetectors equipped with filters 530/40 (blue line, green fluorescence: PNA-FITC) and 613/20 (blue line, red fluorescence: PI). All parameters were visualized on a logarithmic scale. At least 10,000 events corresponding to spermatozoa were collected. Data were processed using Weasel v3.4 (http://www.frankbattye.com.au/Weasel/, accessed on 15 April 2022).

### 2.6. Artificial Insemination

The semen doses were distributed among the farms belonging to the Asturian Holstein Breeders’ Association (ASCOL) from 1999 to 2018. This insemination program is aimed at producing calves with a high beef yield. Therefore, CBA-SERIDA produces AI doses from ASEAMO’s bulls, which are stored in the germplasm bank (also serving as a genetic safeguard for these bulls), and these doses are sent to ASCOL’s farms for insemination. After the voluntary waiting period (VWP), about 60 d for multiparous cows and 90 d for primiparous cows, the animals in oestrus were inseminated following standard protocols by trained veterinarians. Animals showing no signs of oestrus were synchronized using a hormonal protocol, Ovsynch [17]. ASCOL collected the fertility data (56-day non-return to estrous rate, NRR) for each bull, discriminating among heifers (first AI) and cows.

### 2.7. Statistical Analysis

The statistical analyses were carried out with the R statistical package [18]. Descriptive results are shown as medians and 1st and 3rd quartiles of the data distribution The effect of freezing, the relationship between the quality variables, and the clustering with the fertility data were studied using linear mixed-effects models (using the protocol step or the cluster as a fixed effect and the bull as a random effect). The relationship between fresh and post-thawing variables, and with NRR, was studied by Pearson correlations, with *p* < 0.01 (due to the high number of correlations, to reduce the possibility of type-I errors). Bulls were clustered according to initial and post-thawing quality and freezability ((post-thawing-quality–fresh quality)/average quality) using an agglomerative hierarchical clustering algorithm with Ward’s method. The optimal number of clusters in each case was estimated using a set of algorithms provided by the NbClust package [19].

## 3. Results

### 3.1. General Characteristics of the “Asturiana de la Montaña” Semen Samples

The descriptive statistics of the AM fresh semen are shown in Table 1. The population of four adult bulls produced semen samples with a median of 7028 × 10^6^ spermatozoa per ejaculate. Half of the samples (interquartile range) were within (5,10) × 10^9^ spermatozoa. The laboratory analyses (Figure 1 and Appendix A) showed that most samples used for freezing had high motility and few abnormalities. After thawing, most variables changed significantly (*p* < 0.001, *p* = 0.005 for LIN), except WOB (Figure 1h) and the proportion of cytoplasmic droplets (Figure 1j). However, the change amount was different among the variables. The effect sizes (mean ± SEM of the change) for each variable were: MOT −53.0 ± 0.9, PROG −49.6 ± 1.0, VCL 24.2 ± 3.0, VAP −18.1 ± 3.1, VSL −21.7 ± 1.9, LIN −1.9 ± 0.6, STR 4.1 ± 0.6, WOB −1.4 ± 1.0, abnormal forms 11.9 ± 0.3 (reported mostly as increasing bent midpieces), cytoplasmic droplets −0.1 ± 0.1, intact acrosomes 16.6 ± 0.7, HOS test −20.2 ± 0.6, and viability −15.5 ± 1.1.

NRR data were available from 33 bulls for 27 179 inseminations. Average fertility (mean ± SD) was 51.7% ± 7.0 (quartiles 47.1 and 56.3). Heifers showed higher fertility than cows, with 58.4% ± 7.7 vs. 43.9 ± 9.2, respectively (*p* < 0.001).

The bull’s age slightly but significantly influenced a few variables (the year was also initially included in the models, but its effect was confounded with the bull effect). The ejaculate volume increased with a slope of 0.79 ± 0.32 mL per year (*p* = 0.022). Some fresh sperm quality parameters, namely the acrosomal damage and viability, decreased with age (−0.53 ± 0.22, *p* = 0.020; −1.57 ± 0.75, *p* = 0.030). Age was not related to any variable in the post-thawing evaluation but affected the freezability regarding the acrosomal damage (0.13 ± 0.05, *p* = 0.021). This last observation was possibly due to the effect on fresh samples rather than a real effect on sperm freezability.

### 3.2. Correlation of Fresh Semen Quality with Post-Thawing Variables

Table 2 and Table 3 show the Pearson correlation coefficients and their significance. Semen production (volume, concentration, and total spermatozoa) did not correlate with post-thawing parameters. Total motility, HOS test reactivity, and viability in the fresh semen positively correlated with the total and progressive motility, HOS test, and viability post-thawing, and negatively with intact acrosomes (*p* < 0.001, overall, stronger correlations for MOT). Progressive motility negatively correlated with the sperm velocities after thawing, and LIN and WOB weakly correlated with post-thawing VSL (*p* < 0.01). These low correlations should be confirmed in a more extensive study since currently, they are unlikely to have a biological significance.

### 3.3. Effects of Fresh and Post-Thawed Sperm Quality and Freezability on NRR

The fresh semen concentration negatively influenced the NRR (slopes of −0.004 ± 0.002, *p* = 0.035 for heifers; −0.008 ± 0.002, *p* < 0.001, for cows). No influence was found for the ejaculate volume or the mass motility measured from the ejaculate. Considering the CASA assessment, sperm velocities negatively affected NRR for heifers (VCL: −0.17 ± 0.07, *p* = 0.028; VAP: −0.15 ± 0.06, *p* = 0.015; VSL: −0.16 ± 0.06, *p* = 0.011; LIN: −0.39 ± 0.13, *p* = 0.005; STR: −0.75 ± 0.29, *p* = 0.011; WOB: −0.43 ± 0.16, *p* = 0.006).

The effects of the post-thawed semen characteristics and freezability on NRR were significant only for cows for VAP (0.04 ± 0.02, *p* = 0.031), VSL (0.09 ± 0.04, *p* = 0.010), and WOB (0.21 ± 0.07; *p* = 0.002); LIN was significant both for heifers (0.39 ± 0.11, *p* < 0.001) and cows (0.51 ± 0.13, *p* < 0.001). Considering the post-thawing recovery as freezability, the velocities affected NRR in cows (VCL: −9.92 ± 3.74, *p* = 0.027; VAP: 4.21 ± 1.87, *p* = 0.013; VSL: 6.83 ± 2.45, *p* = 0.004), and LIN and WOB in heifers (30.39 ± 5.49, *p* < 0.001; 12.38 ± 4.44, *p* = 0.006) and cows (24.68 ± 7.31, *p* < 0.001; 17.40 ± 5.19, *p* = 0.001).

### 3.4. Clustering of Bulls according to Post-Thawed Quality and Freezability

The clustering of the bulls according to their post-thawing quality produced two clusters (Figure 2). Cluster 1 showed larger total motility, linearity, viability, and, especially, velocity. Cluster 2 grouped bulls with larger post-thawing progressivity. When clustered according to freezability (Figure 3), we obtained three clusters: Cluster 1 grouped bulls with almost no velocity or linearity changes, whereas progressivity was better preserved within Cluster 2. Cluster 3 grouped bulls with lower overall freezability, with the greatest losses in total motility and viability. However, NRR (total, heifers or cows) did not significantly change among post-thawing quality or freezability clusters.

## 4. Discussion

This study defines the main characteristics of the fresh and thawed semen of the “Asturiana de la Montaña” breed from a group of selected bulls. This report not only contributes to the knowledge of the semen characteristics of this breed, but we also report fertility after applying the doses stored at the cryobank and explore the relationship between sperm quality and AI fertility.

Cryobanks are an excellent option for the long-term storage and dissemination of genetics [20,21]. They have helped prevent the loss of genetic diversity in autochthonous breeds and encouraged breeders to enter AI programs [22]. In the case of the AM breed, the collaboration of the Principado de Asturias and the local breeders’ association (ASEAMO) has enabled the preservation of semen doses from selected bulls, applying them in the Holstein farms afterward. Nevertheless, these banks must be backed up by reliable protocols and on good fertility after the application of the frozen semen. To achieve that, it is essential to characterize the reproductive characteristics of each breed. Many authors have shown that semen characteristics and freezability vary among breeds [23,24]. More importantly, current protocols for freezing bull semen are based on commercial breeds [25]; therefore, they should be adapted to indigenous breeds, with a different genetic background.

AM semen has not been characterized previously considering fresh and post-thawed sperm quality and the relationship among quality variables and with field fertility. Thus, our first objective was to compile available data on fresh semen evaluation. However, there is a caveat: Only semen passing the initial assessment was included in the database. Therefore, our results do not represent the whole breed since we do not have information from rejected ejaculates. Considering standard procedures such as those established by the Society for Theriogenology for considering a bull a Satisfactory Potential Breeder [26,27], the semen quality of all AM bulls could be regarded as of good quality (above 60% progressive motility, and 70% normal morphology, including cytoplasmic droplets). Moreover, sperm production assured the production of a high number of doses per ejaculate beyond the needs of the cryobank for most cases. These results are similar to those reported for the related “Asturiana de los Valles” breed [16].

Sperm quality in the thawed semen decreased compared to fresh samples, with less than one-quarter of the samples achieving a “fair” score according to their progressive motility (more than 30%), with a concomitant increase of abnormal forms and acrosomal damage. Compared to other breeds, Zhang et al. [28], using Swedish Red and White bulls, reported that nearly 80% of the thawed samples contained >70% of spermatozoa with linear movement. Morrell et al. [24], studying several commercial breeds, indicated an average progressive motility of 47% for the bottom 10% of bulls (as the fertility index score). Our results suggest that AM semen could be more susceptible to freezing/thawing, not only compared to the commercial breeds but also the related autochthonous breed “Asturiana de los Valles.” After using a similar protocol using the BIOXcell extender [16], these authors reported higher total and progressive motility. This confirms that genetics and management could influence semen characteristics, and ART could be adapted for specific breeds.

However, fertility results after AI using the thawed doses were close to those reported in commercial breeds. For instance, Zhang et al. [28] showed NRR within (53%, 73%), and Oliveira et al. [29] obtained a fertility of 58% (Angus bulls and Nelore cows). We found that heifers presented better fertility than cows, in line with results from other breeds [30,31]. These results suggest that the AI protocol could compensate for lower post-thawing semen quality. A high number of spermatozoa inseminated, adequate synchronization and management protocols, and experienced personnel could explain the good fertility reported in this study. These results also suggest that cryopreservation might exert compensable damage to the spermatozoa [32], as found in a previous study assessing semen frozen with the Biociphos and BIOXcell extenders [13].

Therefore, it is unsurprising that there were few significant relationships between post-thawing sperm quality or freezability and field fertility. Whereas some CASA parameters showed some association, the effects were low, with LIN standing out. This is interesting since LIN has not been cited as a good predictor of bull fertility. Instead, sperm velocity (VCL or VAP) [33,34] has been positively related to NRR after AI. Nevertheless, Farrell et al. [35] included it in multivariate models with other CASA variables, and Zhang et al. [28] found a positive correlation between linear motility patterns and NRR. Since we carried out a retrospective study based on previous AI databases with limited data (no information on female age or technician, for instance), we could not account for the many variables influencing fertility [36]. Moreover, other authors have insisted that individual variables cannot be relied upon for predicting fertility [37]. When considering these results, we must be careful since CASA output heavily depends both on the analysis setup (magnification, chamber, etc.) and software, and minimal changes can heavily modify results (e.g., just changing the frames per second with all other factors fixed [38]).

Nevertheless, it should be desirable to increase sperm quality after thawing to improve the efficiency of the semen bank. As expected for an autochthonous breed with no selection for sperm freezability, semen from many bulls showed a low resilience to cryopreservation. Another problem is the high variability between bulls (large interquartile ranges for many variables, especially post-thawing). The lack of correlations between sperm quality before and after freezing, except for some particular variable pairs (mainly related to total and progressive motility and membrane condition), is possibly due to this between-bull variability. Whereas sperm quality could be enough for obtaining acceptable results for most bulls, applying different AI protocols or other ART (e.g., AI-MOET) might be compromised for some bulls. Indeed, using low-dose AI or after sperm sexing is a valuable strategy for increasing the efficiency of sperm cryobanks [39,40]. Future studies on the described cryobank or others might include field trials with limited sperm per dose, enhancing the ability to discriminate among high- and low-fertility bulls [41] and analyses mimicking the events the spermatozoa undergo post-AI [42].

The lack of differences in NRR between bull clusters after grouping by post-thaw quality or freezability seems to be related to these kinds of limitations in both assessing the sperm doses and relying on standard AI databases. Whereas clustering could enable identifying bulls and aggregation patterns associated with freezability characteristics, the present study could not find a link with field fertility. Enhanced data could make better use of this technique and other statistical approaches (such as supervised clustering strategies [43,44]).

## 5. Conclusions

To sum up, the present study contributes to sperm quality data before and after cryopreservation, which helps to characterize the sperm cryobank for the “Asturiana de la Montaña” breed. This information not only contributes to improving the conservation strategy for this breed, but it could also help manage other endangered cattle breeds. Whereas the field fertility is good enough at the moment, the post-thawing sperm quality could be improved, at least for some bulls. Therefore, future studies might focus on adapting current cryopreservation protocols for this breed and maybe specific individuals.

## Figures and Tables

**Figure 1 animals-13-01402-f001:**
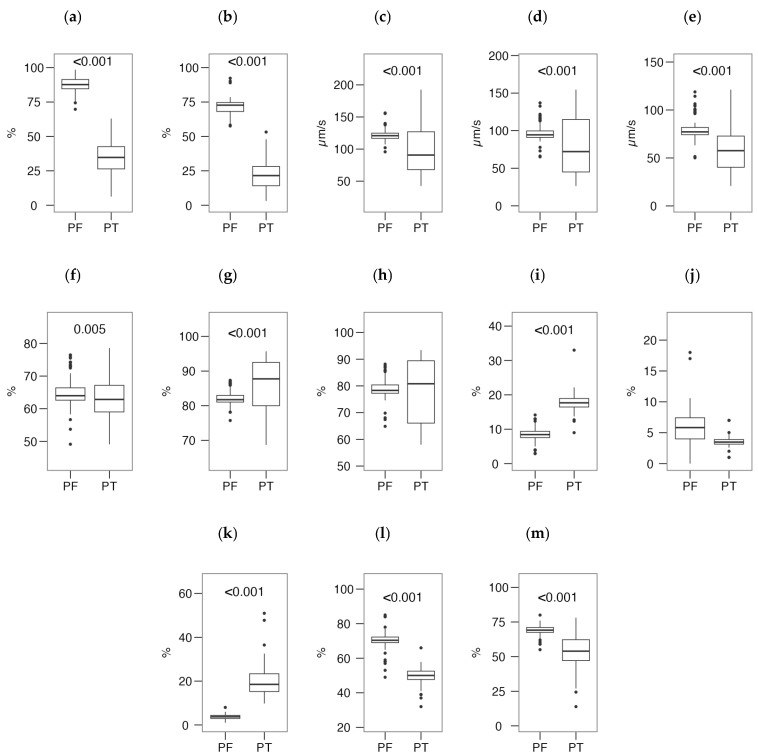
Descriptive statistics from the pre-freezing (PF) and post-thawing (PT) semen analysis (40 Asturiana de la Montaña bulls, 102 ejaculates). Box-plots show 1st and 3rd quartiles of the distribution (box limits), median (inner line), upper and lower hinges of the distribution (vertical lines, extreme observations within 1.5 times the interquartile limits), and outliers (dots, observations outside the upper/lower hinges). *p* values for the effect of the freezing/thawing process are shown on the top when *p* ≤ 0.05. (**a**) Total motility. (**b**) Progressive motility. (**c**) VCL. (**d**) VAP. (**e**) VSL. (**f**) LIN. (**g**) STR. (**h**) WOB. (**i**) Abnormal forms. (**j**) Cytoplasmic droplets. (**k**) Damaged acrosomes. (**l**) HOST. (**m**) Viability.

**Figure 2 animals-13-01402-f002:**
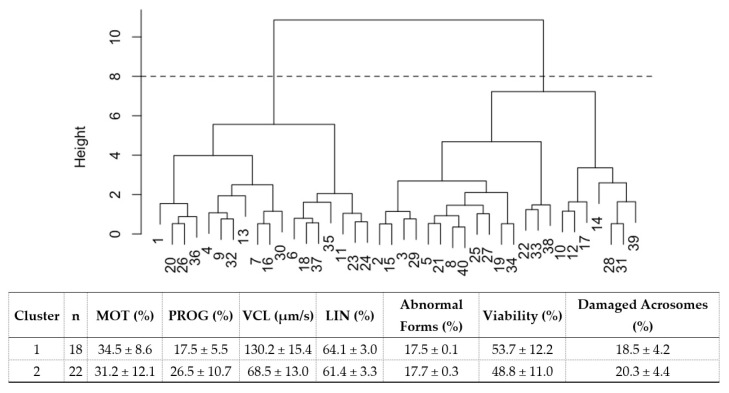
Clustering of the bulls (numbers) according to the post-thawing semen quality parameters. Total motility, VCL, STR, and viability were selected as the most informative and uncorrelated variables for the clustering. The table shows medians ± MAD (median absolute deviation) for each group.

**Figure 3 animals-13-01402-f003:**
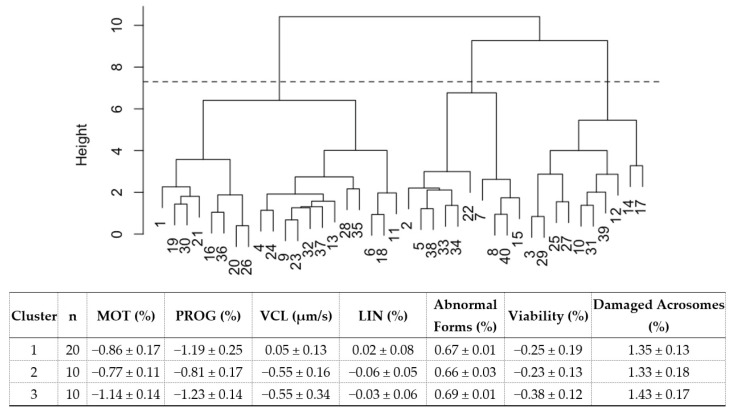
Clustering of the bulls (numbers) according to semen freezability (difference of post-thawing and fresh quality, normalized by their average; negative values indicate a decrease after thawing, positive values indicate an increase, whereas values around 0 indicate no change). Total motility, progressivity, VCL, WOB, and damaged acrosomes were selected as the most informative and uncorrelated variables for this clustering. The table shows medians ± MAD (median absolute deviation) for each group.

**Table 1 animals-13-01402-t001:** Descriptive statistics for the routine evaluation of the ejaculates in the AI center (40 Asturiana de la Montaña bulls, 102 ejaculates).

Variable	Min.	1st Q.	Median	3rd Q.	Max.
Volume (mL)	2	4	5	6	8.5
Concentration (×10^6^ mL^−1^)	687	1199.2	1423.5	1727.8	2404
Total spermatozoa (mill.)	2061	4949	7027.5	9612	19,227
Mass motility (0–5)	4	5	5	5	5

**Table 2 animals-13-01402-t002:** Correlation analysis (Pearson r) between fresh and post-thawing parameters for the Asturiana de la Montaña bulls (cont. to Table 3).

	Thawed							
Fresh	MOT	PROG	VCL	VAP	VSL	LIN	STR	WOB
Volume	−0.04	−0.06	0.09	0.08	0.07	−0.06	−0.05	0.02
Concentration	0.05	0.00	0.04	0.00	−0.06	−0.25 ·	−0.09	−0.04
Total spermatozoa	0.01	−0.04	0.08	0.05	0.01	−0.19	−0.09	−0.01
MOT	0.75 ***	0.57 ***	0.09	0.11	0.13	0.20 ·	−0.05	0.14
PROG	−0.21 ·	−0.13	−0.41 ***	−0.27 **	−0.30 **	0.19	0.02	0.09
VCL	−0.05	−0.10	0.14	0.15	0.18	0.14	−0.09	0.15
VAP	−0.03	−0.11	0.19	0.19	0.22 ·	0.16	−0.12	0.19
VSL	−0.02	−0.09	0.20 ·	0.20 ·	0.23 ·	0.16	−0.11	0.19
LIN	0.01	−0.07	0.23 ·	0.22 ·	0.26 **	0.18	−0.11	0.19
STR	0.04	−0.01	0.18	0.17	0.21 ·	0.15	−0.06	0.14
WOB	−0.02	−0.10	0.23 ·	0.23 ·	0.27 **	0.18	−0.13	0.20 ·
Abnormal forms	−0.02	0.00	−0.06	−0.05	−0.05	0.01	0.04	−0.03
Cytoplasmic droplets	0.02	0.00	−0.09	−0.05	−0.06	0.12	−0.03	0.06
Damaged acrosomes	0.04	0.06	−0.08	−0.06	−0.05	0.10	0.06	0.00
HOS test	0.36 ***	0.32 **	−0.04	−0.05	−0.04	−0.01	0.03	−0.03
Viability	0.53 ***	0.42 ***	0.01	0.04	0.06	0.21 ·	−0.03	0.13

· *p* < 0.05; ** *p* < 0.01; *** *p* < 0.001; *p* values considered significant for *p* < 0.01 for compensating for multiple correlations.

**Table 3 animals-13-01402-t003:** Correlation analysis (Pearson r) between fresh and post-thawing parameters for the Asturiana de la Montaña bulls (continued from Table 2).

	Thawed				
Fresh	Abnormal Forms	Cytoplasmic Droplets	Damaged Acrosomes	HOS Test	Viability
Volume	0.05	−0.02	0.06	−0.11	−0.06
Concentration	0.00	0.06	−0.04	−0.02	0.07
Total spermatozoa	0.03	0.02	0.02	−0.08	0.01
MOT	−0.02	0.05	−0.79 ***	0.62 ***	0.99 ***
PROG	0.19	0.13	−0.05	−0.31 **	−0.02
VCL	0.24 ·	0.20 ·	−0.01	−0.15	−0.03
VAP	0.12	0.14	−0.03	−0.11	−0.03
VSL	0.11	0.14	−0.02	−0.10	−0.04
LIN	−0.02	0.08	−0.04	−0.04	−0.05
STR	0.03	0.12	0.02	0.00	−0.09
WOB	−0.05	0.05	−0.06	−0.07	−0.03
Abnormal forms	−0.17	0.08	−0.03	−0.10	0.01
Cytoplasmic droplets	0.15	0.10	0.01	0.06	0.06
Damaged acrosomes	0.03	0.23 ·	−0.02	0.02	0.05
HOS test	0.13	0.01	−0.36 ***	0.34 ***	0.48 ***
Viability	0.17	−0.03	−0.60 ***	0.51 ***	0.81 ***

· *p* < 0.05; ** *p* < 0.01; *** *p* < 0.001; *p* values considered significant for *p* < 0.01 for compensating for multiple correlations.

## Data Availability

Data is available upon request to the corresponding author.

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
