# Peer review of "Characterization of the Germplasm Bank for the Spanish Autochthonous Bull Breed “Asturiana de la Montaña”"

_animals, 2023, doi:10.3390/ani13081402_

Round 1
Reviewer 1 Report
General Comments:
The authors provide an interesting perspective on the current freezing techniques for this specific breed in Spain and how that might need to be adjusted to improve pregnancy success with using assisted reproductive techniques.
Specific Comments:
Line 39- What is deep-frozen semen? All semen used in assisted reproductive technologies is deep-frozen in LN2.
Line 49- “World” needs to be lowercased
All the subheadings are the same numerically, and will need to be changed to fit each section appropriately.
Lines 98-99- Why were two different types of extenders used? Was each extender used in the same bull? How many bulls received each different type of extender?
Lines 145-147- Were the estrous synchronization protocols the same between cows and heifers, or were there different protocols used?
Statistical analysis section:
How was pre- vs. post-thaw semen analyzed? This is not fully explained in this section.
If there were significant differences between pre and post breeding, then this data would be well suited as a graph.
Were the two types of extenders evaluated?
Line 167- Is this significant difference from pre- vs. post-thaw analysis? If so, how was this data analyzed?
Lines 193-194- This needs to be moved to the statistical analysis section.
Lines 293-294- Why were two different extenders used if previous research demonstrated differences in cryopreservation rates? Was this information taken into account for this project?
Line 315- The statement should read before and after freezing.
Line 320- There shouldn’t be a . in “spe.rm”
Author Response
The authors appreciate the reviewers’ comments and the editorial help. We expect that the manuscript has attained the required quality for publication in Animals.
General Comments:
The authors provide an interesting perspective on the current freezing techniques for this specific breed in Spain and how that might need to be adjusted to improve pregnancy success with using assisted reproductive techniques.
> Thank you for your kind comment. We have tried to make the manuscript clearer following the indications.
Specific Comments:
Line 39- What is deep-frozen semen? All semen used in assisted reproductive technologies is deep-frozen in LN2.
> Deep-frozen is used as synonym for frozen and stored in LN2. We agree with the reviewer that it could be more specific to write just “frozen in LN2.” We indicated that for distinguishing from refrigerated storage (limited use in cattle, but widespread for other species such as swine and sheep).
Line 49- “World” needs to be lowercased
> Thank you, done.
All the subheadings are the same numerically, and will need to be changed to fit each section appropriately.
> Thank you. This is automatic formatting from the MDPI submission system. We have corrected that.
Lines 98-99- Why were two different types of extenders used? Was each extender used in the same bull? How many bulls received each different type of extender?
> The reason for using Biociphos and then BIOXcell was the discontinuation of the first one by the manufacturer, IMV. From the 40 bulls, 15 were frozen with Biociphos and 25 with BIOXcell. We have added this information to the manuscript.
Lines 145-147- Were the estrous synchronization protocols the same between cows and heifers, or were there different protocols used?
> Thank you for pointing out this detail. We accidentally omitted this relevant information, now added.
Statistical analysis section:
How was pre- vs. post-thaw semen analyzed? This is not fully explained in this section.
> We agree. It was explained by line 157, at the end of the section, but possibly not in a good position. We have moved it by the beginning now, and we have expanded the description (we used linear mixed-effects models).
If there were significant differences between pre and post breeding, then this data would be well suited as a graph.
> Thank you. We preferred the tables, since we wanted to offer a good glimpse of the situation of the germplasm bank to the reader. However, we agree that plots could be more intuitive. We have included them in the main manuscript, and we have moved the tables to supplementary material, therefore providing the full information without repeating data.
Were the two types of extenders evaluated?
> Samples frozen with the two extenders were included out of neccesity, since this is a retrospective study using semen doses already stored in the semen bank for a long period (information now included). Since we wanted to offer a full description of the germplasm bank situation, we decided to include all these samples.
Line 167- Is this significant difference from pre- vs. post-thaw analysis? If so, how was this data analyzed?
> Yes. As indicated in that line, P<0.001, except P=0.005 for LIN and P>0.05 for WOB and cytoplasmic droplets. As indicated above, the statistics description has been clarified, and the addition of plots shows these results more directly.
Lines 193-194- This needs to be moved to the statistical analysis section.
> We agree, done.
Lines 293-294- Why were two different extenders used if previous research demonstrated differences in cryopreservation rates? Was this information taken into account for this project?
> Yes, as indicated above, these samples are still stored in the germplasm bank. Despite some effect of the extender on some quality variables, we decided to include all doses in the study, since it offered a full description of the bank, and our previous testing demonstrated no effect on field fertility.
Line 315- The statement should read before and after freezing.
> Thank you, corrected.
Line 320- There shouldn’t be a . in “spe.rm”
> Thank you, corrected.
Reviewer 2 Report
Authors conducted a retrospective study about the seminal quality of seminal doses of Asturiana de la Montana breed in Spain. These samples were all collected in a reproduction centre with adequate protocols. However, just high genetic bulls are included and all frozen doses were from suitable bulls with a good BBSE evaluation. Therefore, it is expectable not to have differences further than "bull" ones.
Moreover, they included fertility outputs, but there is too much information missing to understand these results.
In general, it is an interesting study, but describing the seminal quality of different processed doses in this centre it is not novel at all. They should be very similar (all with fertilizing capacity enough), and just show "bull" differences. However, the fertility outcomes they link with these doses is very interesting, but, they have to improve these results and be more clear in order to publish this data.
Some other comments:
- missing simple summary
- ln 33-42. it is, as well, in beef cattle. Altough the AI cows per herd is quite lower, the AI is the main tool to increase genetic merit of beef hers, and the main used. As the breed you are studying is mainly for beef production, support it with this idea.
- ln 71. "many authors" but you do not include references
- ln 65-79. the objectives are not properly expressed. I would encourage authors to rewrite this paragraph
- ln 81. I think this section should be 2.1
- ln 85. which legislation? references
- ln 89. missing (
- it is a retrospective study. years of seminal sampling and freezing?
- ln 108. 2.2, please take care of the format throughout the manuscript. I would not point this out again
- ln 143. More information is needed about how they do the AI. It is not the same when performing AI after oestrus vs. FTAI, even in FTAI protocol could also interfere. Year is important as well, or farm. This section is very interesting, but it has to be very controled to report fertility outputs
- ln 163. you said (ln 90) 40 bulls...
- ln 169-172 is interesting but is difficult to understand like this, I would place it into a table. Maybe you can merge fresh, frozen and differences data in a huge table to see at once all the results.
Why do you give this data as mean +- sd if you are using median and interquartile ranges throughout the reuslts? I guess you used median and interquartile range because data did not follow a normal, but you did not explain it before.
- ln 183. same comment as before.
- ln 185. It is interesting about age, but what about year?
- table 4. In my humble opinion, it the rho value is not < |0.4|, I would not consider it as a correlation.
- ln 206. But you said you freeze seminal doses at a constant concentration...???
- ln 216-218. you are giving results for cows and heifers, but I could not see reproductive outputs of that AIs
- Clusters. If there is no reproductive benefit or loss (ln231), what is the aim os this clusters?
- ln 253. you said ASCOL previously, is there any difference?
- ln 264-266. Completely right. In fact, as you are a reproduction centre, I think the bulls you have there are very selected and with a very high genetical outcome. Thus, it would not represent the whole bulls that are in the field preserving this breed. Is thi right? Or do they all AI with this doses and there is no natural mating?
- ln 286-287. I still thinking that these results are not properly expressed, therefore, I cannot review this statement.
- ln 304. nor protocol, and some other important variables
- ln. 320-323. any information about inbreeding reduction??
Author Response
Authors conducted a retrospective study about the seminal quality of seminal doses of Asturiana de la Montana breed in Spain. These samples were all collected in a reproduction centre with adequate protocols. However, just high genetic bulls are included and all frozen doses were from suitable bulls with a good BBSE evaluation. Therefore, it is expectable not to have differences further than "bull" ones.
Moreover, they included fertility outputs, but there is too much information missing to understand these results.
> We thank the reviewer’s comments. We want to clarify that, as indicated in the manuscript, AM is a local, authochthonous breed. Therefore, it is not subject to the same selection programs than commercial breeds, and there is a much lower homogeneity among males. Males producing low-quality semen are not immediately removed, since at this moment the objective of the SERIDA and the breeders’ associations is to expand and preserve the genetic variability of the breed. Whereas there is an interest to improve the beef potential of AM (mainly to produce beef by crossing with commercial, more intensively reared dairy breeds), this selection is not as intense as in commercial breeds.
In general, it is an interesting study, but describing the seminal quality of different processed doses in this centre it is not novel at all. They should be very similar (all with fertilizing capacity enough), and just show "bull" differences. However, the fertility outcomes they link with these doses is very interesting, but, they have to improve these results and be more clear in order to publish this data.
> Thank you for your comments. This study's novelty focuses on a non-commercial, autochthonous, and endangered breed and the usefulness and viability of this germplasm bank. As indicated in the text, little is known about the reproductive specifics of this breed. Contrarily to commercial breeds, the management and breeding of endangered breeds have different objectives. While achieving a benefit for the farmers, there are cultural and social objectives, and the primary goal is to preserve the breed, assuring its genetic diversity and protecting it against future threats, and to enable crossing with dairy breeds for producing calves for beef.
We have revised the study following the reviewer’s comments to improve the clarity and enhancing the manuscript.
Some other comments:
- missing simple summary
> Thank you for pointing it out. We submitted the manuscript as free format, and in the process we forgot to add the simple summary (now included). We also have detected some minor mistakes caused by the conversion process to Animals’ format and corrected them.
- ln 33-42. it is, as well, in beef cattle. Altough the AI cows per herd is quite lower, the AI is the main tool to increase genetic merit of beef hers, and the main used. As the breed you are studying is mainly for beef production, support it with this idea.
> We agree. This first part was a bit confusing. We have ammended it following the suggestions.
- ln 71. "many authors" but you do not include references
> Thank you. Indeed, this is confusing, since we added the cites by the end of the following sentence. We have ammended the text.
- ln 65-79. the objectives are not properly expressed. I would encourage authors to rewrite this paragraph
> We appreciate your constructive criticism. We have followed your advice to make the objectives more evident.
- ln 81. I think this section should be 2.1
> Thank you. Since we submitted as “free format,” section numbering was all the same. Corrected.
- ln 85. which legislation? references
> We refer to law providing protection for farm animals, added.
- ln 89. missing (
> Thank you, corrected.
- it is a retrospective study. years of seminal sampling and freezing?
> Yes, this is an evaluation of the cryobank for assessing its past performance and detect threats to its future viability for preserving this local, endangered breed, and for enabling crossings with dairy breeds (futher explanation in the AI section). We have added information about the years to the manuscript in section 2.1 and in section 2.6.
- ln 108. 2.2, please take care of the format throughout the manuscript. I would not point this out again
> We have revised the manuscript. As indicated above, the problems with the format arose because we sent the manuscript as free format (allowed by the journal) and the MDPI-formatted version was produced afterwards (not available for checking before confirming the submission).
- ln 143. More information is needed about how they do the AI. It is not the same when performing AI after oestrus vs. FTAI, even in FTAI protocol could also interfere. Year is important as well, or farm. This section is very interesting, but it has to be very controled to report fertility outputs
> The reviewer is right, this section lacked some relevant information. We have to add that the objective of the study is not specifically to carry out an AI trial (the data was provided by the associations’ activity). Currently, the breed status has improved, but AI is not used as a routine technique in this breed yet. This is a rustic, local breed extensively reared in mountainous areas, and natural mating is the norm. Nevertheless, there is an incentive to apply germplasm banking and AI to this breed, to enable crossing with dairy commercial breeds to improve the beef yield of the calves.
- ln 163. you said (ln 90) 40 bulls...
> Thank you for pointing it out. Corrected.
- ln 169-172 is interesting but is difficult to understand like this, I would place it into a table. Maybe you can merge fresh, frozen and differences data in a huge table to see at once all the results.
> Thank you for the observation. The other reviewer suggested using plots, therefore we have presented the results in the Fig. 1 and the tables are maintained in the supplementary material.
Why do you give this data as mean +- sd if you are using median and interquartile ranges throughout the reuslts? I guess you used median and interquartile range because data did not follow a normal, but you did not explain it before.
> These are effect sizes. As indicated in the text, these are mean±SEM of the change of the variable (or difference, as provided by the linear mixed-effects models). That is, the average amount the variable changes from before to after freezing/thawing.
- ln 183. same comment as before.
> In this case, the data came from a larger dataset and other studies present this kind of results by using SD. We considered that it could be convenient showing both the mean±SD and the interquartile range limits.
- ln 185. It is interesting about age, but what about year?
> This is an opportune question. We explored different, more complex models, but the structure of the data was not appropriate to study the year (confounded with the bull). We have added an explanation by that line.
- table 4. In my humble opinion, it the rho value is not < |0.4|, I would not consider it as a correlation.
> We agree that such a small correlation is not biologically meaningfull, even if significant. We have added a sentence in this sense.
- ln 206. But you said you freeze seminal doses at a constant concentration...???
> We refer to the fresh semen (added “fresh” as clarification). Since sperm concentration and other parameters of the fresh sample could be a proxy of spermatogenesis activity and other reproductive variables, we considered including these parameters in an association study.
- ln 216-218. you are giving results for cows and heifers, but I could not see reproductive outputs of that AIs
> They are described in “3.1. General characteristics of the ‘Asturiana de la Montaña’ semen samples.”
- Clusters. If there is no reproductive benefit or loss (ln231), what is the aim os this clusters?
> The purpose of this technique was the same as any other technique, to test the hypothesis that bull groups defined by this technique could be associated to NRR (that is, that they could differ on NRR). We found some well separated groups in terms of sperm quality and freezability, but they did not reflect on NRR differences.
- ln 253. you said ASCOL previously, is there any difference?
> ASCOL is the Heifer’s breeders association, and ASEAMO is the AM breeders’. After adding more information for explaining the AI, we believe this is clearer now.
- ln 264-266. Completely right. In fact, as you are a reproduction centre, I think the bulls you have there are very selected and with a very high genetical outcome. Thus, it would not represent the whole bulls that are in the field preserving this breed. Is thi right? Or do they all AI with this doses and there is no natural mating?
> As explained above, this is not the case with this breed. The bulls are not strictly selected and the selection program pursues maintaining the genetic diversity while expanding the breed (still considered in risk). This is why we refer to the lack of representativity due to the semen quality assessment, not because of strict bull selection. Whereas some bulls are rejected due to very poor sperm quality, this procedure is not as stringent as for commercial breeds seeking the maximum performance.
As it is now explained in the text, AI is carried out in dairy farms working with commercial breeds. In the field, natural mating is the norm.
- ln 286-287. I still thinking that these results are not properly expressed, therefore, I cannot review this statement.
> We are not sure about this comment. We interpret that the fertility results reported by ASCOL as NRR are comparable with some previous studies, and therefore we can state with some confidence that the fertility of the semen doses in the SERIDA cryobank for AM is acceptable.
- ln 304. nor protocol, and some other important variables
> Included now. We hope that this information help to clarify this part of the discussion.
- ln. 320-323. any information about inbreeding reduction??
> As far as we know, only a study has been carried out in this topic, in 2007. This is in Spanish, and available here: https://marcalyc.redalyc.org/toc.oa?id=495&numero=10289
The authors conclude that the breed had a moderate selection pressure and a relatively moderate inbreeding increase. These authors highlight that the intervention in the breed management, including pedigree records completion, reduced the growth of the average kinship, but that deeper interventions were needed. We believe that this trend could have improved in the last years, but no new studies are available.
Round 2
Reviewer 2 Report
Authors answered every one of my suggestions. However, I have two main issues that makes me think that the experimental design is not appropriate at all.
1) In total there are 40 samples taken with almost 20 years of difference (from 1999 to 2018). Moreover, it is not balanced the extender used. Therefore, I think it is difficult to compare this results with this high time lapse.
2) The fertility outputs are not comparable, because there are several factors that could interfere, beyond using one or another seminal sample.
Author Response
Thank you for your comments. The reviewer is correct that the data and design are inappropriate for these inferences. Whereas this is true, we want to highlight that the purpose and objectives of the study are not those to describe the germplasm bank and to assess its viability for preserving the breed (that is, maintaining a genetic reserve of significant males) and for the practical utility for the breeder's associations (crossing with commercial dairy breeds for beef production).
In this context, the time span is instead an asset than a liability. As mentioned, a germplasm bank's purpose is to preserve the genetics of relevant animals, eventually helping breed management and conservation. Including "old" doses has enabled us to confirm the viability of those doses in case they are needed. As explained in the manuscript, these doses were frozen with the extender available then. Differences between extenders were considered in the other study, confirming a better performance of BIOXcell, but still finding a post-thawing quality good enough to enable the use of these doses.
Similarly, for the fertility data, our aim was not to predict it from the quality data or make other inferences. We assessed the fertility data from the practical application of these doses in thousands of AI. Whereas it would have been very interesting to have additional data regarding the different factors affecting AI outcomes, for our purposes, the NRR allows us to confirm the germplasm bank's viability and compare it with similar data from other studies. We have discussed these topics in the manuscript and concluded that our data support the usability of the bank doses.